# OpenReview forum: "Audio Flamingo 3: Advancing Audio Intelligence with Fully Open Large Audio Language Models"
_NeurIPS.cc/2025/Conference — NeurIPS 2025 spotlight_

### Official Review · Reviewer_P65i · 2025-06-29

**Clarity:** 4
**Significance:** 4
**Originality:** 3
**Rating:** 5
**Confidence:** 3

**Summary:**

The authors present a large audio-language model for general audio understanding across speech, sound, and music. The core architectural contribution is AF-Whisper, a unified audio encoder that can process various audio types within a single module. This encoder is combined with a Qwen-2.5-7B LLM and a streaming TTS module to form the overall architecture. The authors also introduce a multi-stage training recipe using a self-curated dataset across multiple tasks, including general/long-form audio understanding, conversational audio, audio reasoning tasks, and voice-in/voice-out scenarios. The model is evaluated on over 20 standard audio benchmarks and demonstrates strong performance. The authors also commit to open-sourcing the code, checkpoints, and data (with partial data already provided in the submission) upon paper acceptance.

**Questions:**

Please see the weakness above; the reviewer has posed several actionable questions.

**Ethical Concerns:**

["NO or VERY MINOR ethics concerns only"]

**Final Justification:**

Thanks for the clarification. The reviewer was satisfied with additional explanations, clarifications, and experiments. Therefore, the score was maintained as Accept (5.0).

**Limitations:**

yes

**Quality:**

3

**Strengths And Weaknesses:**

Strengths
1. Commitment to open-source: If fully released, the code, model, and dataset would be a valuable contribution to the community given that there are limited fully open-source ones in this field.
2. Unified audio encoder (AF-Whisper): The introduction of a single encoder that works well on diverse audio inputs simplifies the overall system design and is a great contribution.
3. Strong performance with 7B backbone: The model achieves competitive results across a broad range of benchmarks despite using a relatively compact 7B language model.

Weaknesses
1. Experimental Setup
	- Missing baselines (e.g., GPT-4o-audio): The authors note (line 279) that GPT-4o and Gemini were not included due to rate limits. While understandable, it would still be valuable to report results on a small subset of tasks, especially since GPT-4o-audio is currently a key reference model in the field.
	- Audio Captioning (Table 2): This task appears to show the smallest gains. It would help to understand why results are relatively weak here. Also, why aren’t results for Audio Flamingo 2 on AudioCaps reported? More recent LLM-based evaluation metrics [1, 2] could improve the analysis here.
	- Evaluation model bias: Many results in Table 2 (reasoning tasks) and Table 3 are evaluated using GPT-4o. Can the authors add comparisons using other LLMs to rule out evaluation bias?

2. Other Points
	- Ablation details (Table 4): (1) It seems that AF-Whisper is based on Whisper-Large-v3 (1.5B). Were parameter sizes controlled for in the ablation when comparing dual-encoder setups (e.g., with CLAP)? (2) The line “+ w/o AF-Whisper” is unclear. Does it mean with or without the encoder?
	- Licensing issue: The authors mention that licensing information is provided in Appendix G, but the appendix currently lacks the promised details, especially for those that the authors collected.
	- Context length and token cost: It would be helpful to specify the maximum context length supported by the model (is it inherited from Qwen-2.5-7B as 128K?) and how many tokens per second are required for different audio types.

[1] CLAIR-A: Leveraging Large Language Models to Judge Audio Captions

[2] MACE: Leveraging Audio for Evaluating Audio Captioning Systems

---

> ### Author Rebuttal · Authors · 2025-07-28
>
> Dear Reviewer P65i,
>
> We thank you for your thorough review and constructive feedback. We're grateful for your recommendation to accept our paper. We have tried to address each of your concerns point by point.
>
> ------
>
> ### Weaknesses
>
> > Missing baselines (e.g., GPT-4o-audio): The authors note (line 279) that GPT-4o and Gemini were not included due to rate limits.
>
> **Ans.** Thank you for the question! We would like to clarify a potential misunderstanding: we do compare AF3 with GPT-4o-audio and all Gemini versions across all benchmarks—classification, captioning, and QA—except for ASR benchmarks. The reason for this exception is that ASR datasets are large and often exceed API rate limits for these closed models. That said, to address this concern during the rebuttal, we ran limited-scale evaluations on four ASR benchmarks, and present the results below:
>
> | Model | Libri (clean) | SPGI | Vox |
> |----------|----------|----------|----------|
> | GPT-4o-audio    | 2.90     | 4.91    | 8.42     |
> | Gemini 2.5 Pro   |  2.04   | 4.30     | 9.79     |
> | Audio Flamingo 3    | **1.51**    | **1.86**     | **5.66**     |
>
> We will include these results and clarifications in the final version of the paper.
>
> > Audio Captioning (Table 2): This task appears to show the smallest gains. It would help to understand why results are relatively weak here.
>
> **Ans.** Thank you for the question! AF3 shows relatively smaller gains on AudioCaps and Clotho because of inherent limitations in current captioning benchmarks and evaluation protocols. Below, we outline key factors:
>
> - Captioning is a perception task—not a reasoning task: Audio captioning primarily evaluates descriptive understanding, not complex reasoning. As a result, foundation models like AF3, which are optimized for reasoning and general audio intelligence, show modest improvements in this task compared to reasoning-heavy benchmarks.
>
> - Limitations of current evaluation metrics: Metrics like BLEU, CIDEr, and SPICE are biased toward style matching, not semantic correctness. For example: Clotho may label a sound as “a forest,” -- while AudioCaps might describe it as “animals among rustling leaves.” -- Both are correct, but style-biased metrics reward only the phrasing closer to the reference.
>
> Since AF3 is trained on diverse real-world audio, its captions are often semantically accurate but stylistically different, leading to under-rewarded performance. *We intentionally did not fine-tune to match benchmark-specific styles, as that would constitute benchmark hacking (e.g., training with prompts like “in Clotho style”).*
>
> > Why no Audio Flamingo 2 scores on AudioCaps?
>
> **Ans.** AF2 was evaluated zero-shot and was not trained on AudioCaps. It reports a baseline CIDEr score of 0.58. In contrast, models like Pengi are trained on AudioCaps, making them fairer points of comparison in Table 2.
>
> > LLM-based evaluation metrics for Audio Captioning
> Thank you for pointing us to [1, 2]. We agree that these can provide better insights into caption quality. Below, we provide results using MACE [2]:
>
> | Model | AudioCaps | Clotho |
> |----------|----------|----------|
> | Gemini 2.5 Pro   |  0.59    | 0.56     |
> | Audio Flamingo 2    | 0.47     | 0.42     |
> | Qwen2.5 Omni    | 0.51     | 0.55    |
> | Audio Flamingo 3    | **0.59**     | **0.60**    |
>
> > Evaluation model bias
>
> **Ans.** Thank you for the question! To address concerns around evaluation model bias, we evaluated AF3 and baselines using additional LLMs beyond GPT-4o. Specifically, we report results, using the same prompt, with Qwen2.5-7B-Instruct:
>
> | Model | LibriSQA | LongAudioBench |
> |----------|----------|----------|
> | Gemini 2.5 Pro   | 8.0    | 68.5     |
> | Audio Flamingo 3    | 8.1  | 79.1    |
>
> And Llama-3.3-70B-Instruct below:
>
> | Model | LibriSQA | LongAudioBench |
> |----------|----------|----------|
> | Gemini 2.5 Pro   | 9.2    | 72.3     |
> | Audio Flamingo 3    | 9.2  | 78.6    |
>
> > Ablation details (Table 4): (1) It seems that AF-Whisper is based on Whisper-Large-v3 (1.5B). Were parameter sizes controlled for in the ablation when comparing dual-encoder setups (e.g., with CLAP)? (2) The line “+ w/o AF-Whisper” is unclear. Does it mean with or without the encoder?
>
> **Ans.** Thank you for the question!
>
> - On parameter control in dual-encoder setups: We did not perform strict parameter control in the ablation. Our intent was to demonstrate that a dual-encoder setup (e.g., CLAP + Whisper-Large-v3) inherently increases model size and leads to less stable training, without a proportional performance gain. To the best of our knowledge, there are no standard methods in the literature for parameter-matched comparisons in dual-encoder architectures with heterogeneous backbones. Our goal is to highlight the advantages of a unified encoder, as introduced via AF-Whisper, which handles speech, sound, and music in a single, efficient module.
>
> - On the line “+w/o AF-Whisper” in Table 4: This refers to the dual-encoder baseline, where we use CLAP for non-speech audio (sound/music) and Whisper-Large-v3 for speech. This setup is used to compare against the AF-Whisper unified encoder, which processes all audio modalities. We will clarify this in the final version of the paper.
>
> > Licensing issue
>
> We apologize for the oversight—Appendix G will be updated in the final version to include full details on licensing. All datasets introduced in this work will be released under a non-commercial, research-only license. This includes the datasets we collected and curated specifically for training AF3. This is due to the licensing terms and data sources used; all assets will be distributed under a research-only license upon acceptance. (Please note that by “fully open,” we mean that the model weights, training data, and code will be publicly released, along with transparent documentation of the training methodology. We do want to strictly abide by the licenses and terms of the datasets provided by the community for training, for our release, most of which are research-only.)
>
> > Context length and token cost
>
> Thank you for the question! AF-Whisper encodes a 30-second audio segment into 750 tokens, and our model is trained with a context length of 16K tokens, corresponding to up to 10 minutes of audio, as stated in the paper.
>
> However, as correctly noted, the model inherits a maximum context length of 128K tokens from Qwen-2.5-7B. That said, we do not guarantee performance beyond 16K, as: i) The model was not explicitly trained or optimized for longer contexts. ii) There are currently no available benchmarks to evaluate such long audio comprehensively. We will clarify these details in the final version of the paper.

---

### Official Review · Reviewer_7pz4 · 2025-07-02

**Clarity:** 4
**Significance:** 3
**Originality:** 3
**Rating:** 5
**Confidence:** 4

**Summary:**

The paper proposes Audio Flamingo 3 (AF3), an audio-language model that unifies speech, sounds, and music understanding within a single framework. AF3 combines a AF-Whisper encoder with a lightweight “think” module to support tasks from transcription and captioning to expert-level reasoning, multi-turn dialogue, and voice-to-voice interaction. The authors curate four big datasets (AudioSkills-XL, LongAudio-XL, AF-Think, and AF-Chat) and use a five-stage curriculum learning to progressively enhance encoder alignment, reasoning, context extension, and conversational fine-tuning. Results show taht AF3 achieves state-of-the-art performance on over 20 audio understanding and reasoning tasks.

**Questions:**

1. Can the model start reasoning without explicit prompting?
2. What are AF3’s latency (RTF) and total parameter count? It doesn’t seem to run in real time, right?
3. How do the cut-offs when stitching segments affect performance? Have you tested different overlap or boundary settings?
4. The voice-chat sounds robotic—have you explored ways to capture natural prosody and interruptions?
5. What are the architecture details of the "audio adaptor" and the "streaming TTS"?

**Ethical Concerns:**

["NO or VERY MINOR ethics concerns only"]

**Final Justification:**

I thank the authors for their detailed response and for addressing my concerns point by point. The clarifications on reasoning behavior, latency, overlap handling, TTS design, and the distinction between foundational and thinking checkpoints were very helpful.

Some issues remain such as English-only training, high computational requirements and the lack of natural prosody in generated speech. Nonetheless, I find the overall contribution significant: the paper demonstrates a strong, integrated audio-language framework with state-of-the-art performance across a wide range of tasks.

In summary, despite some limitations, this is a technically solid and impactful paper that advances the field of general-purpose audio-language models, and I maintain my original score.

**Limitations:**

Although section 7 mentions limitations in the title, a dedicated limitations section is missing. The authors should include such a section and provide a systematic overview of all known constraints - including language coverage, compute requirements, pipeline latency, and prosody limitations - alongside an explicit discussion of potential societal impacts (e.g. demographic biases, privacy risks).

**Quality:**

3

**Strengths And Weaknesses:**

Strengths

1. Model and data will become publicly available.
2. A single encoder (AF-Whisper) handles speech, sounds, and music.
3. State-of-the-art performance on 20+ benchmarks

Weaknesses

1. English-only training.
2. High computational cost. Training and inference need substantial compute.
3. The cascaded voice-chat pipeline strips out prosodic nuances - so conversations feel flat and robotic, with no natural interruptions or emphasis.
4. Method-wise the contributions are incremental.

---

> ### Author Rebuttal · Authors · 2025-07-28
>
> Dear Reviewer 7pz4,
>
> We thank you for your thorough review and constructive feedback. We're grateful for your recommendation to accept our paper and address each of your concerns point by point.
>
> -----
>
> ### Questions:
>
> > Can the model start reasoning without explicit prompting?
>
> **Ans.** Thank you for the question! No, AF3 does not reason without explicit prompting. *This is by design*—our training methodology and the AF-Think dataset were created to support a novel on-demand thinking paradigm. Unlike domains such as math or coding, many audio tasks (e.g., ASR, captioning) do not inherently require reasoning. Therefore, the model is trained to reason only when explicitly instructed to do so, e.g., via a thinking prompt like “Please think and reason before you respond.”
>
> We also take this opportunity to clarify the distinction between our two checkpoints:
>
> - AF3 foundational checkpoint (after Stage 3): Suited for core audio tasks such as classification, captioning, and ASR.
>
> - AF3 thinking checkpoint (after Stage 3.5): Trained with an additional LoRA adapter using AF-Think, targeted at tasks requiring deliberate reasoning and long audio comprehension.
>
> Importantly, the thinking checkpoint also behaves like the foundational model when the reasoning prompt is absent. However, due to its exposure to reasoning-style supervision, it may be biased toward that style and perform slightly worse on basic tasks—this is why we keep the checkpoints separate (as also noted in Table 7 in the Appendix, where Stage 3 data is not fully reused in Stage 3.5). **We will open-source both checkpoints upon acceptance.**
>
> > What are AF3’s latency (RTF) and total parameter count? It doesn’t seem to run in real time, right?
>
> **Ans.** AF3's total parameter count is 8.2B (Qwen2 LLM + Whisper Encoder), similar to most existing LALMs (Qwen2-Audio, GAMA, LTU, etc.) AF3's latency (RTF) is 0.14 computed on the test set of LibriSpeech-clean.
>
> Real-time performance depends heavily on the inference infrastructure and GPU used. While AF3 supports real-time low-latency streaming outputs via TTS (see Appendix Section I for details), it does not yet support streaming audio inputs, which, together with the model size, may limit its real-time applicability, particularly on edge devices (real-time can still be achieved on larger GPU settings). *That said, most existing LALMs also lack real-time input support on edge devices.* Our current focus is on achieving general audio intelligence through scaling data and model capacity. Real-time performance is an important future goal, and we plan to release smaller, efficient versions of the next AF versions to support deployment on edge devices.
>
> > How do the cut-offs when stitching segments affect performance? Have you tested different overlap or boundary settings?
>
> **Ans.** Thank you for the question. *Our initial small-scale experiments showed a drop in performance when segments had boundary overlap.* Unfortunately, we do not have full-scale results across different boundary settings, as we were constrained by compute resources. However, since our windowing strategy is inspired by Audio Flamingo 2, we’d like to share results from AF2 comparing different overlap settings:
>
> | Model | CMM | MMAU-Sound | MuchoMusic |
> |----------|----------|----------|----------|
> | 4 sec overlap   | 55.0     |  45.6   | 31.0     |
> | 2 sec overlap   | 73.0     | 57.8     | 43.8     |
> | no overlap   | 82.0     | 65.1    | 56.5     |
>
> We observed that overlapping segments in long and structural audios (like music) may confuse the model, causing it to misinterpret boundaries and lose coherence. This likely happens because, in the early stages of training, the model sees mostly short, single-segment audio, learning to treat each segment as a coherent unit. The concept of overlapping segments is only introduced later (during long audio fine-tuning), making it harder for the model to generalize. Additionally, for audios like music where the structure develops over time, the overlapping model confuses the model by losing coherence.
>
> > The voice-chat sounds robotic—have you explored ways to capture natural prosody and interruptions?
>
> **Ans.** Thank you for the question. Capturing natural prosody and interruptions was beyond the scope of Audio Flamingo 3, as it requires training a dedicated TTS model on expressive speech data (or just simply integrating external TTS systems into AF3 as a cascaded setup).
>
> Our current TTS implementation serves as a functional demonstration, showing that low-latency voice-out is achievable with LALMs in a simple and unified manner. *With appropriate data, it is relatively simple to achieve this (although most of such data is not open and would defy our open claim if we used)*. As noted in the caption of Table 1, voice-out is not the core novelty of AF3—but our streaming TTS implementation itself is novel, as described in Section I of the Appendix.
>
> It's also worth noting that existing voice-to-voice systems (e.g., duplex-style models) primarily focus on voice interaction, not on audio intelligence, which we define as holistic perception, understanding, and reasoning over audio. Achieving audio intelligence with natural prosody remains an open challenge and is a key priority for future versions of Audio Flamingo.
>
> > What are the architecture details of the "audio adaptor" and the "streaming TTS"?
>
> **Ans.** Thank you for the question. The audio adaptor consists of a two-layer MLP with a GeLU activation in between. We will include these architectural details in the revised version of the paper. Details of the streaming TTS setup are provided in Appendix I, Figure 10. It outlines our lightweight, low-latency design used for generating audio in real-time from model outputs.
>
> ------
>
> ### Weaknesses
>
> > English-only training.
>
> **Ans.** We fully acknowledge this limitation and have noted it in the Limitations section of our paper. Expanding to multilingual speech understanding is a key goal for the next version of Audio Flamingo. Interestingly, despite not including any multilingual speech data during training, AF3 exhibits some emergent multilingual capabilities, which we attribute to the inherent strengths of Whisper and Qwen2—both of which are multilingual models.
>
> > High computational cost. Training and inference need substantial compute.
>
> **Ans.** Thank you for the question. We acknowledge the high computational cost, which is common to most foundational models. AF3 is designed as a general-purpose, intelligent audio-language model capable of handling complex reasoning over diverse audio inputs. That said, AF3 is compute-efficient relative to its performance:
>
> - Our intelligent data curation and training design allow AF3 to outperform more powerful models like Qwen2-Audio [1], Qwen2-Omni [2], and LTU-AS, while using significantly less data.
>
> - As a result, AF3 requires less compute for training compared to these models.
>
> At inference time, AF3 has a comparable cost to other leading models in the literature, including Qwen2-Audio, Qwen2-Omni, and LTU-AS.
>
> > The cascaded voice-chat pipeline strips out prosodic nuances - so conversations feel flat and robotic, with no natural interruptions or emphasis.
>
> **Ans.** Thank you for the comment. We addressed related concerns earlier in the rebuttal, but we’d like to add a specific clarification about the cascaded voice-chat pipeline.
>
> While end-to-end voice-to-voice models (e.g., using codec inputs/outputs) may better preserve prosodic nuances, implementing such a system in AF3 presents challenges. In our initial experiments, training AF3 with codec-in/out led to catastrophic forgetting, causing the model to lose its core audio intelligence—its ability to holistically perceive and reason over diverse audio types.
>
> We acknowledge that models like Moshi [3] handle natural prosody well, but they are not designed for audio intelligence. These models focus on translating textual conversations into speech using LLM capabilities, without reasoning over the audio content itself (e.g., sounds, background noise, speaker variation). *Bringing together natural prosody and deep audio understanding remains an open challenge, and a key research direction for future versions of Audio Flamingo.*
>
> > Method-wise the contributions are incremental.
>
> **Ans.** Thank you for the comment! We would like to take this chance to highlight some contributions: The core novelty of our work lies in unlocking and demonstrating advanced capabilities in audio-language models—including long audio comprehension, multi-audio and multi-turn dialogues, on-demand reasoning, and emergent thinking—capabilities that were largely absent from the literature, and how to enable them remained unknown in open-access models, which did not expose or document such behaviors, just open-sourced weights. As a result, how to effectively elicit and train these capabilities remained an open question, which our work directly addresses.
>
> Our goal was to build a strong, general-purpose audio-language foundation model, not only by innovating in isolated components, but by designing the entire system holistically—from novel data creation paradigms and stage-wise training strategies to the unified encoder architecture -- thereby finding the answers to the unknown and revealing it to the community. While we acknowledge that individual elements may not be entirely novel in isolation, that was not our aim. We believe that audio intelligence still benefits greatly from scaling both data and parameters, and we demonstrate that doing so meaningfully enhances performance.
>
> ------
>
> > Although section 7 mentions limitations in the title, a dedicated limitations section is missing.
>
> **Ans.** Thank you for the comment. We will add this to the final version of our paper, describing the limitations broadly.
>
> ------
> ### References
> [1] https://arxiv.org/abs/2407.10759.
> [2] https://arxiv.org/abs/2503.20215.
> [3] https://arxiv.org/abs/2410.00037.

---

### Official Review · Reviewer_o9GS · 2025-07-02

**Clarity:** 4
**Significance:** 4
**Originality:** 4
**Rating:** 6
**Confidence:** 5

**Summary:**

This paper introduces Audio Flamingo 3 (AF3), a fully open large audio-language model (LALM) capable of comprehensive reasoning and understanding across speech, sound, and music. AF3 advances the field by introducing several key innovations:

- AF-Whisper, a unified audio encoder trained via a novel captioning-based representation strategy across three audio modalities.

- Support for on-demand thinking (inspired by chain-of-thought), multi-turn multi-audio chat, voice-to-voice interaction, and long-context audio reasoning (up to 10 minutes).

- Four newly curated high-quality datasets---AudioSkills-XL, LongAudio-XL, AF-Think, and AF-Chat---designed to improve task-specific reasoning.

- A five-stage curriculum-based training framework, progressively increasing context length and task complexity.

AF3 and its chat variant (AF3-Chat) achieve state-of-the-art performance across over 20 benchmarks spanning audio QA, reasoning, ASR, captioning, and multi-modal dialogue. The model is positioned as the most capable and transparent open-source LALM to date.

**Questions:**

1. How does AF3 handle real-world noisy or overlapping audio in multi-source environments?

2. Were any metrics (e.g., latency or memory) collected on the streaming TTS system in deployment settings?

3. Is there any evidence of LLM hallucination or bias in AF-Think or AF-Chat, and how are such cases handled?

4. Are there plans for multilingual extensions or use of Whisper’s multilingual capability in AF-Whisper?

**Ethical Concerns:**

["NO or VERY MINOR ethics concerns only"]

**Final Justification:**

The work is technically solid with comprehensive evaluation; All my concerns have been appropriately addressed.

**Limitations:**

AF3's performance on noisy or real-world conversational audio is not deeply analyzed despite model scope suggesting applicability.

**Quality:**

4

**Strengths And Weaknesses:**

**Strengths:**

- Excellent empirical results across a wide range of tasks, including long-context QA, sound/music reasoning, and speech understanding.

- Significant open-source release, including trained models, code, datasets (13M+ audio-caption pairs), and demo interface.

- Well-grounded training methodology, including dataset-specific prompting, skill-specific QA generation, and multi-modal data balancing as detailed in the appendix.

- Streaming TTS design is thoroughly described in the appendix, with a detailed architectural diagram, codec specs, training pipeline, and latency optimization.

- Rich appendix clarifies dataset composition (e.g., Table 7 with hours and pairs per dataset), training hyperparameters (Table 8), and prompting strategies (Figures 11–46), enhancing reproducibility.

- Evaluation on MMAR and AIR-Bench (Appendix Table 9) further strengthens the claim of general-purpose audio reasoning ability.

**Weaknesses:**

- Closed LLMs used for synthetic data generation (GPT-4.1, Gemini) reduce full replicability despite an open model release.

- Only English supported, limiting multilingual utility, though data and tools could support extensions.

- AF-Think's effect on model robustness is described but ablations are shallow.

---

> ### Author Rebuttal · Authors · 2025-07-28
>
> Dear Reviewer o9GS,
>
> We thank you for your thorough review and constructive feedback. We're grateful for your recommendation to strongly accept our paper. We have tried to address each of your concerns point by point.
>
> -----
> ### Weaknesses
>
> > Closed LLMs used for synthetic data generation (GPT-4.1, Gemini) reduce full replicability despite an open model release.
>
> **Ans.** Thank you for the question. To ensure full reproducibility, we will release all datasets used to train the model—including AudioSkills-XL, LongAudio-XL, AF-Think, and AF-Chat—alongside the model checkpoints, training code, and data processing scripts, as stated in the paper. While we used closed LLMs (GPT-4.1, Gemini) for some synthetic data, we are committed to releasing the final generated datasets to ensure end-to-end applicability.
>
> > Only English supported, limiting multilingual utility, though data and tools could support extensions.
> > Are there plans for multilingual extensions or use of Whisper’s multilingual capability in AF-Whisper?
>
> **Ans.** Thank you for the suggestion. We fully acknowledge this limitation and have noted it in the Limitations section of our paper. Expanding to multilingual speech understanding is a key goal for the next version of Audio Flamingo. Interestingly, despite not including any multilingual speech data during training, AF3 exhibits some emergent multilingual capabilities, which we attribute to the inherent strengths of Whisper and Qwen2—both of which are multilingual models. This further motivates our plan to incorporate explicit multilingual supervision in future work.
>
> Multilingual extensions would only require adding multilingual data to the mixture, and we plan to follow the same process as AF-Whisper and AF3 training with appropriate weights to multilingual data in the mixture.
>
> > AF-Think's effect on model robustness is described but ablations are shallow.
>
> **Ans.** Thank you for the question. While we included a basic ablation in the main paper, we provide additional clarification below to address the depth of AF-Think’s effect on model robustness, answering the most important questions:
>
> - *Does the model always generate long, CoT-style answers after AF-Think training?*: No. The model only engages in reasoning when explicitly prompted (e.g., with the thinking prompt shown in Fig. 3). Without this, it defaults to concise, task-appropriate responses. This demonstrates robust control over reasoning behavior.
>
> - *Does training on AF-Think hurt foundational performance?* No. We evaluate Audio Flamingo 3 on MMAU-test without the thinking prompt after Stage 3.5 (i.e., post AF-Think training) and observe no drop in performance, achieving scores of 73.11 and 73.0 on MMAU-test and MMAU-test-mini, respectively.
>
> - *Does thinking mode always improve performance?* Not always. For simpler tasks like ASR or sound event classification, thinking mode may not help—and we do not expect it to (e.g, we maintain an almost similar score of 78.9 in NSynth Instrument Classification and 1.92  on SPGI ASR). The thinking mode is designed to enhance tasks that require multi-step reasoning, such as those measured by MMAU and MMAR, where it shows clear gains.
>
> ------
>
> ### Questions
>
> > How does AF3 handle real-world noisy or overlapping audio in multi-source environments?
>
> **Ans.** AF3 is able to handle real-world noisy and overlapping audio to *some extent*. We attribute this to training examples from our Speech-in-Sound Captions and QA data, which include speech with naturally occurring background noise, interruptions, and mild cross-talk from AudioSet and YouTube8M.
>
> However, we do not expect AF3 to perform well in extreme noise or dense overlapping speech scenarios, as it was not explicitly trained on such data. Prior work has shown that robust performance in these settings requires specialized training data and objectives [1]. **Handling multi-speaker, multi-source, and noisy audio is a key focus for the next version of Audio Flamingo.** We plan to include datasets such as Chime, LibriheavyMix, etc., in training and evaluate on benchmarks designed for these challenging real-world conditions.
>
> > Were any metrics (e.g., latency or memory) collected on the streaming TTS system in deployment settings?
>
> **Ans.** Thank you for the question. Yes, we report latency metrics in Table 3.
>
> Specifically, for a 10-second audio generation on an A100 GPU:
> - Text-to-audio token generation: 5.94 seconds
> - Waveform synthesis: 0.02 seconds
> - Total time: 6.68 seconds for a 10-second audio clip
>
> In a streaming TTS setup, we observe:
> - Time-to-first-token: 0.15 seconds
> - Inter-token latency: 0.06 seconds (both including synthesis)
>
> These metrics demonstrate that AF3-Chat can support low-latency, real-time text-to-speech generation suitable for deployment settings.
>
>
> > Is there any evidence of LLM hallucination or bias in AF-Think or AF-Chat, and how are such cases handled?
>
> **Ans.** Yes, we did observe instances of LLM hallucination and bias during the creation of AF-Think and AF-Chat. To mitigate this, we implemented the following safeguards:
>
> - In AF-Think, some responses in the thought process referred incorrectly to elements like “the caption” or “meta-data,” which were not part of the input. We used keyword-based filtering to remove such hallucinated examples.
>
> - In AF-Chat, early generations occasionally reflected hallucinations stemming from inaccurate captions or transcripts. We addressed this through iterative caption refinement, which significantly improved data quality.
>
> We will add these details to the final version of our paper. That said, we acknowledge that the datasets are not entirely noise-free. *For future versions of Audio Flamingo, a key focus will be on developing and integrating automated data filtering techniques to further reduce hallucinations and improve supervision quality at scale.*
>
>
> ------
>
> ### References
>
> [1] He, Xinlu, and Jacob Whitehill. "Survey of End-to-End Multi-Speaker Automatic Speech Recognition for Monaural Audio." arXiv preprint arXiv:2505.10975 (2025).

---

> > ### Comment · Reviewer_o9GS · 2025-08-06
> > **Response to the Authors**
> >
> > Thank you for the detailed rebuttal; it effectively addressed all my concerns.
> >
> > Your plan to release the curated datasets is an excellent solution for the reproducibility issue. The clarification on AF-Think's behavior was very insightful and should be added to the final paper. I also appreciate your transparency regarding the model's limitations (noise, latency, hallucinations) and your clear plans for future work.
> >
> > My assessment is unchanged. This is a Strong Accept.

---

### Official Review · Reviewer_mZSh · 2025-07-02

**Clarity:** 3
**Significance:** 3
**Originality:** 2
**Rating:** 5
**Confidence:** 3

**Summary:**

This paper proposes a third version of Audio Flamingo, abbreviated as AF3, a large audio language model with new capabilities and datasets to enable it. In particular, capabilities added over AF2 include tackling speech through a unified encoder (AF-Whisper) for speech, music and sound, multi-turn multi audio chat and CoT style thinking. The paper proposes better, larger versions of AudioSkills(-XL) and LongAudio(-XL), and new datasets AF-Chat and AF-think for the respective capabilities. Evaluation is carried out over several benchmarks demonstrating superior performance. AF-Whisper encoder and AudioSkills data contributions are also analyzed to show improved performance.

**Questions:**

Please also see weaknesses

- Could the authors clarify how the +Think version is different from AF3. In particular,  they mention it is AF3 with thinking prompts; is it different from those incorporated in the AF-think dataset and training curriculum for CoT style thinking? I also ask because, in Table 2, +Think is marked in gray (closed source), but the authors mention in the abstract that they will open source “..all our code, data, and checkpoints upon paper acceptance”. If there is indeed a discrepancy, it will be good to clarify up front what will and won't be released.

- Something I am curious about: on the demo webpage the authors show "Emergent Audio Understanding" results. What would happen if the instructions are more general and do not specify one or both of the events in the audio - it one just asked the model to say what is surprising, atypical or unlikely (respectively) about the events in this audio?

**Ethical Concerns:**

["NO or VERY MINOR ethics concerns only"]

**Final Justification:**

After the rebuttal, I have a more nuanced understanding of the contributions. It also addressed some of my questions around limitations and model release plan. Increasing my score to 5.

**Limitations:**

yes

**Quality:**

3

**Strengths And Weaknesses:**

Strengths
- In my view, datasets are the main contribution of this paper, as they are also the main drivers of the capabilities of AF3. In particular, speech-in-sound QA for audio skills and LongSpeech QA for long audio seem to be very timely contributions for audio reasoning benchmarks.
- AF-whisper is proposed as a unified encoder that does better than the dual encoder approach with one for sounds/music and another for speech.
- Extensive evaluation over multiple benchmarks
- Fully open source model and datasets to help advance research on the topic.
- The paper is well written and easy to follow.

Weaknesses
- The paper has incremental novelty in technical approach, limited to an additional step in curriculum learning over AF2 and the unified encoder.
- It would have been interesting to analyze when and how the model fails after training with proposed well-curated datasets.

---

> ### Author Rebuttal · Authors · 2025-07-28
>
> Dear Reviewer mZSh,
>
> We thank you for your thorough review and constructive feedback. We're grateful for your recommendation to accept our paper. We have tried to address each of your concerns point by point.
>
> ------
> ### Questions:
>
> > Could the authors clarify how the +Think version is different from AF3. In particular, they mention it is AF3 with thinking prompts ....
>
> **Ans.** The *+Think* model refers to the version trained in Stage 3.5 using the AF-Think (and LongAudio XL) dataset, where a) we append a thinking prompt—“Please think and reason before you respond”—to each question in the AQA instances and b) each response has a thought process before arriving at the actual answer to encourage chain-of-thought (CoT) -type reasoning. This differs from the Stage 3 AF3 model, which is trained without such prompting and does not reason by default (the version trained in Stage 3.5 is also able to process longer audios, up to 10 minutes).
>
> This design is intentional. Unlike domains such as math or code, most audio understanding tasks do not inherently require structured reasoning. We therefore propose a novel *on-demand thinking* mechanism, where the model engages in reasoning only when explicitly prompted by the user. This setup provides greater flexibility and better aligns with real-world usage, where reasoning is not always needed.
>
> **Regarding model release**: *We (will) open-source all models*, including +Think, as stated in the abstract. The use of gray shading for +Think in Table 2 was unintentional. They are actually different shades of grey. Thank you for pointing out that there is confusion, and we will change this in the final version to avoid confusion with closed-source models.
>
> > Something I am curious about: on the demo webpage the authors show "Emergent Audio Understanding" results. .....
>
> **Ans.** Thank you for the thoughtful question. In the first demo example, we tested the model with a more general prompt: “What is atypical about the events in the audio?”—without explicitly naming any events. The model responded: “The dog’s barking is atypical as it is accompanied by music.” This demonstrates that the model can identify salient events and reason about their relationships, even when the prompt is underspecified. It suggests an emergent ability to infer context and highlight surprising or unusual aspects in complex audio scenes.
>
> ------
>
> ### Weaknesses
>
> > The paper has incremental novelty in technical approach, limited to an additional step in curriculum learning over AF2 and the unified encoder.
>
> **Ans.** Thank you for the comment! The paper proposes much more novel findings beyond just a minor curriculum update. We would like to take this chance to highlight the same: The core novelty of our work lies in unlocking and demonstrating advanced capabilities in audio-language models—including long audio comprehension, multi-audio and multi-turn dialogues, on-demand reasoning, and emergent thinking—*capabilities that were largely absent from the literature, and how to enable them remained unknown in open-access models, which did not expose or document such behaviors, just open-sourced weights. As a result, how to effectively elicit and train these capabilities remained an open question, which our work directly addresses.*
>
> Our goal was to build a strong, general-purpose audio-language foundation model, not only by innovating in isolated components, but by designing the entire system holistically—from *novel data creation paradigms* and stage-wise training strategies to the unified encoder architecture -- thereby finding the answers to the unknown and revealing it to the community. While we acknowledge that individual elements may not be entirely novel in isolation, that was not our aim. We believe that audio intelligence still benefits greatly from scaling both data and parameters, and we demonstrate that doing so meaningfully enhances performance.
>
> Going forward, scaling reinforcement learning and reward-driven training will be a key focus to further improve grounding and reasoning capabilities. Importantly, AF3 is fully open—including code, data, and checkpoints—offering the community the first powerful and transparent model for audio understanding and reasoning.
>
> > It would have been interesting to analyze when and how the model fails after training with proposed well-curated datasets.
>
> **Ans.** Thank you for the question. Our proposed training method has undergone extensive ablations, where we developed a 5-stage training recipe that contributed to AF3’s performance. Key factors include: a) Audio windowing, b) Dataset weighting, c) Stage-wise freezing/unfreezing of model components, d) Hyperparameter tuning, e) Unified audio-text encoder design -- these design decisions make AF3 robust. *However, despite these, AF3 has some known limitations:*
>
> - **Training Data Biases**:  In rare stress-test cases, AF3 produces responses that include spurious or memorized tokens unrelated to the input. These cases often stem from biases in the training data and are a known phenomenon in LLM/MLLM fine-tuning [1]. We plan to explore reinforcement learning (e.g., RLHF) to mitigate such issues in future versions.
>
> - **Limited Instruction Robustness**: AF3 tends to perform best on instructions seen during training, and its generalization to novel or out-of-distribution instructions is still limited. Increasing instructional prompt diversity during training is a key direction for improvement.
>
> - **Hallucinations in Long-Form Generation**: While AF3 performs well for short, factual responses, we observe occasional hallucinations in longer, open-ended outputs. This is a known challenge in MLLMs [2]. We aim to improve audio grounding and incorporate retrieval-augmented generation techniques to enhance long-form fidelity.
>
> - **Multi-Turn Dialogue Degradation**:  AF3 maintains coherent multi-turn conversation up to ~8 turns, after which hallucinations may increase. We plan to investigate dialogue-aware training methods to extend robustness over longer interactions.
>
> - **Coverage Gaps in Specific Capabilities**:  AF3 currently underperforms in: i) Multicultural music understanding, ii) Multilingual speech comprehension, iii) Noisy and overlapping speech scenarios. These gaps primarily arise from the lack of such data in the training corpus. Addressing this through targeted additional data is a priority for future releases.
>
> We will add these comprehensively in the revised version of our paper.
>
> ------
> #### References:
>
> [1] https://proceedings.mlr.press/v235/ghosh24a.html.
> [2] https://openreview.net/forum?id=3PRvlT8b1R

---

> > ### Author Response · Authors · 2025-08-05
> > **Request to review the rebuttal**
> >
> > Dear reviewer mZSh,
> >
> > Thank you for taking the time to review our paper. We have addressed your concerns in our submitted response. As the rebuttal period is nearing its conclusion, we kindly request you to review our rebuttal and share any additional comments or concerns you may have. Thank you once again for your valuable feedback!
> >
> > Best,
> > Authors of Submission13594

---

> > ### Comment · Reviewer_mZSh · 2025-08-05
> >
> > Thanks for the clarifications. I have no additional comments.

---

> ### Author Response · Authors · 2025-08-05
> **Response to Official Comment by Reviewer mZSh**
>
> Thank You! That is great to hear! Would be grateful if you could **adjust your scores if you would like to, and also finish the Mandatory Acknowledgment!**
>
> Best,
> Authors of Submission13594

---

> > ### Comment · Reviewer_mZSh · 2025-08-07
> >
> > Sure. Thanks for addressing my concerns. I do have a more nuanced understanding of the contributions. I shall update my score accordingly.

---

> > > ### Author Response · Authors · 2025-08-07
> > > **Thank You!**
> > >
> > > Thank You for the consideration! We are glad we address your concerns! Please let us know if you would like further clarifications, we would be glad to assist!
> > >
> > >
> > > Best,
> > > Authors of Submission 13594

---

### Decision · Program_Chairs · 2025-09-17

**Decision:**

Accept (spotlight)

**Comment:**

This paper presents Audio Flamingo 3 (AF3), a fully open large audio-language model with a unified encoder, new large-scale datasets, and a staged curriculum enabling long-context reasoning, multi-turn/multi-audio dialogue, and voice-to-voice interaction. AF3 achieves state-of-the-art results on 20+ benchmarks and commits to releasing all models, code, and data, making it a significant community resource. While some methodological contributions are incremental and limitations remain (English-only training, reliance on closed LLMs for synthetic data, limited prosody in TTS, and incomplete robustness analysis), the work was judged technically solid, well-evaluated, and impactful. Reviewers’ concerns were addressed convincingly during rebuttal, with clarifications and new experiments provided. Overall, I recommend acceptance (spotlight): the openness, breadth of capabilities, and strong empirical results make AF3 a valuable and timely contribution to audio-language modeling.